# Enhancing Task-oriented Dialogue Systems with Generative Post-processing Networks

**Atsumoto Ohashi**       **Ryuichiro Higashinaka**

Graduate School of Informatics, Nagoya University

ohashi.atsumoto.c0@s.mail.nagoya-u.ac.jp

higashinaka@i.nagoya-u.ac.jp

## Abstract

Recently, post-processing networks (PPNs), which modify the outputs of arbitrary modules including non-differentiable ones in task-oriented dialogue systems, have been proposed. PPNs have successfully improved the dialogue performance by post-processing natural language understanding (NLU), dialogue state tracking (DST), and dialogue policy (Policy) modules with a classification-based approach. However, they cannot be applied to natural language generation (NLG) modules because the post-processing of utterances output by NLG modules requires a generative approach. In this study, we propose a new post-processing component for NLG, generative post-processing networks (GenPPNs). For optimizing GenPPNs via reinforcement learning, the reward function incorporates dialogue act contribution, a new measure to evaluate the contribution of GenPPN-generated utterances with regard to task completion in dialogue. Through simulation and human evaluation experiments based on the MultiWOZ dataset, we confirmed that GenPPNs improve the task completion performance of task-oriented dialogue systems[1].

## 1 Introduction

A typical task-oriented dialogue system has a pipelined structure consisting of four modules (Young et al., 2013; Zhang et al., 2020): natural language understanding (NLU), dialogue state tracking (DST), dialogue policy (Policy), and natural language generation (NLG). Many studies have used reinforcement learning (RL) to improve the task completion performance of an entire pipelined dialogue system by fine-tuning modules directly (Lee et al., 2021; Lin et al., 2021; Chen et al., 2023).

Recently, Ohashi and Higashinaka (2022b) proposed a novel method using Post-Processing Networks (PPNs) that can optimize pipelined dialogue

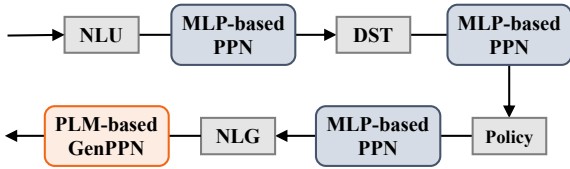

Figure 1: Diagrams of MLP-based PPN for NLU, DST, and Policy modules proposed by Ohashi and Higashinaka (2022b) and PLM-based GenPPN for NLG proposed in this study.

systems that consist of arbitrary modules including non-differentiable ones (e.g., rule-based and Web API-based). Their method uses RL to optimize a neural-based component called a PPN, which modifies the output of each module. PPNs have been applied to three modules, namely NLU, DST, and Policy, and have been shown to improve the task completion performance of various pipelined systems. In PPNs, the post-processing of module outputs is treated as a binary classification task, i.e., adding or removing slots in the output of each module, and this classification task is modeled with a multi-layer perceptron (MLP). This means that the current PPNs could not be applied to NLGs, where the unit of output is not slots but a sequence of tokens, i.e., natural language.

To overcome this limitation, we propose a Pretrained Language Model (PLM)-based Generative Post-processing Network (GenPPN) that can post-process the output of NLGs in a manner similar to conventional PPNs (Figure 1). To optimize GenPPN via commonly used RL frameworks for PLMs (Ziegler et al., 2019; Stiennon et al., 2020), a reward for each utterance at each turn (an *utterance-level* reward) is required. In a task-oriented dialogue, however, a reward indicating success or failure is obtained only at the end of a multi-turn interaction (a *dialogue-level* reward). Therefore, we introduce a dialogue act (DA) contribution for distributing the dialogue-level reward to the utterance-

---

[1]Our code is publicly available at https://github.com/nu-dialogue/GenPPN

level reward. Here, a DA is the meaning representation of the information that NLG converts into an utterance, and DA contribution is a measure of how much the DA of each utterance contributes to the final task completion of the dialogue.

Experiments on the MultiWOZ dataset (Budzianowski et al., 2018) confirm that GenPPNs can improve the task performance of the entire dialogue system, regardless of the architecture of the NLG module. Furthermore, an ablation study reveals that the introduction of DA contribution is effective for learning GenPPNs to improve task completion. The contributions of this study are threefold:

- We propose the generative post-processing network (GenPPN) to modify the output utterances of the NLG module, which was impossible with conventional PPNs.

- We introduce DA contribution to optimize the GenPPN by evaluating the impact of each utterance on dialogue task completion.

- Simulation experiments on the MultiWOZ dataset confirm that the proposed GenPPN improves the task completion performance of the entire dialogue system, regardless of the architecture of the NLG module. We also validated that a GenPPN optimized using simulation is effective in human evaluation experiments.

## 2 Related Work

### 2.1 Optimization of Pipelined Task-oriented Dialogue Systems

Methods have been proposed to optimize the task completion performance of an entire pipelined system using RL. Zhao and Eskenazi (2016) and Li et al. (2017) optimized a Policy network implemented in MLP using the Deep Q-Network algorithm (Mnih et al., 2013) to achieve robustness against errors that occur in real dialogues. Mehri et al. (2019) proposed a method to add additional parameters to NLU, Policy, and NLG and optimize them using RL. By expressing a dialogue state output by a DST as a probability distribution, Lee et al. (2021) made the entire system differentiable and jointly optimized it via RL. A method of fine-tuning an NLU while optimizing a Policy in dialogue simulations was proposed (Lin et al., 2021). Tseng et al. (2021) proposed a domain-adaptive learning

framework that simultaneously optimizes the Policy module of the dialogue system and the user simulator. An RL framework for online DST optimization was also proposed to improve dialogue management performance (Chen et al., 2023).

Instead of training each module, Ohashi and Higashinaka (2022b) proposed a generalized method that optimizes PPNs, classification-based models that modify the outputs of NLU, DST, and Policy, to enhance the task completion performance. In this paper, we propose a new generative-based PPN for post-processing NLG modules, which has not been supported by conventional PPNs.

### 2.2 Natural Language Generation Module for Task-oriented Dialogues

Conventional NLGs for task-oriented dialogues used template-based or rule-based methods (Walker et al., 2002; Stent et al., 2004). Later, data-driven methods using machine learning were proposed (Oh and Rudnicky, 2002; Angeli et al., 2010; Mairesse and Young, 2014) that do not require the cost of template and rule creation.

In recent years, many generative models based on deep learning have been proposed (Wen et al., 2016; Tran and Nguyen, 2017; Su et al., 2018). Wen et al. (2015) proposed an SC-LSTM that controls utterance generation using DA feature vectors and reading gates. SC-GPT (Peng et al., 2020) is the best NLG model of MultiWOZ. It achieves high performance by fine-tuning GPT-2 (Radford et al., 2019) on many task-oriented dialogue datasets such as MultiWOZ (Budzianowski et al., 2018) and the Schema-Guided Dialogue Dataset (Rastogi et al., 2020).

For task-oriented dialogue, it is crucial not only to generate natural utterances via maximum likelihood estimation (MLE) but also to accurately reflect the input DA's content. To achieve this, Balakrishnan et al. (2019) introduced a conditional decoding approach utilizing a tree-shaped semantic representation, enhancing the slot content in generated utterances. Furthermore, Li et al. (2020) offered a method to lower the slot error rate in utterances using an iterative RL framework for slot consistency. Ohashi and Higashinaka (2022a) presented a fine-tuning method for utterance generation that uses a user's NLU model so that the NLU can understand the DA accurately.

In all of the above studies, the optimization was performed at the utterance level using a fixed cor-

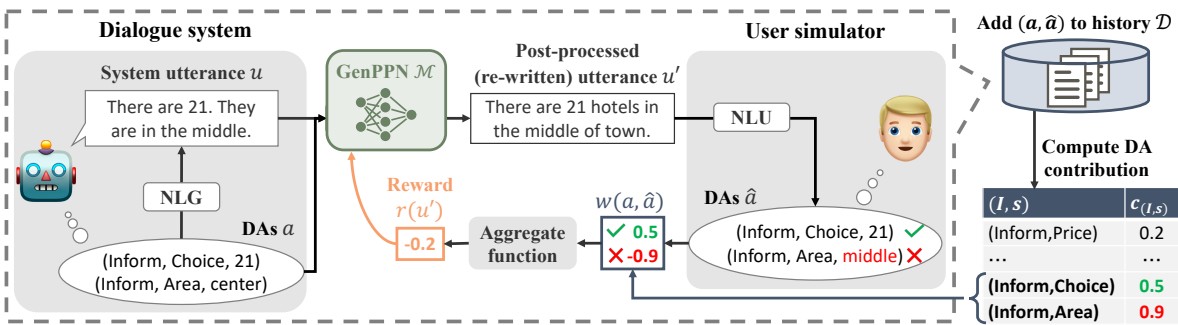

Figure 2: Schematic diagram of GenPPN and how it is optimized. GenPPN $\mathcal{M}$ rewrites utterance $u$ generated by NLG of dialogue system into $u'$. This rewriting of $\mathcal{M}$ is optimized using reward $r(u')$ calculated on the basis of DA contribution $c_{(I,s)}$, so system's ability to complete task improves.

pus of utterances. In this study, we aim to optimize post-processing not only on the utterance level but also on the dialogue level to improve the task completion performance for multi-turn dialogues.

## 3 Method

### 3.1 Overview

Figure 2 shows a schematic diagram of GenPPN and how it is optimized. At the beginning of each interaction, the user is given a user goal consisting of a list of constraints to be informed to the system and information to be obtained from the system through the dialogue. During the dialogue, from the user's utterance at a certain turn $t$, the NLU and DST modules of the system first estimate the current belief state of the user. Then, on the basis of the belief state, the Policy module determines the next actions to be taken by the system as DAs $a_t$. The $a$ takes a structure containing one or more triples of intent $I$, slot $s$, and value $v$:

$$a = \{(I_i, s_i, v_i) \mid i = 1, ..., |a|\} \quad (1)$$

The NLG module maps $a_t$ to the system's utterance $u_t$. Here, GenPPN $\mathcal{M}$ post-processes, i.e., rewrites $u_t$. A prompt $x_t = \text{Prompt}(h_t, a_t, u_t)$ is created from the previous dialogue context $h_t$, $a_t$, and $u_t$, and $\mathcal{M}$ generates a new utterance $u'_t$: $u'_t \sim \mathcal{M}(x_t)$ (see Section 3.2 for details on GenPPN and the prompt format). The user then receives $u'_t$, estimates the system DAs $\hat{a}_t$, and the user's utterance is output for the next turn $t+1$ on the basis of the $\hat{a}_t$ and current status of the user goal. This exchange is repeated until the user goal is achieved or until the maximum number of turns is reached.

Let $T$ be the number of turns at the end of a dialogue. In addition, the dialogue result $e \in \{0, 1\}$ of whether the user goal was achieved is determined,

i.e., 0 means that the task failed, and 1 means that the task succeeded. The objective is to obtain the optimal parameters $\theta^*$ for the GenPPN $\mathcal{M}_\theta$ such that the expected value of $e$ is maximized:

$$\theta^* = \arg\max_\theta \mathbb{E}_{U' \sim \mathcal{M}_\theta(X)} \big[e(U')\big] \quad (2)$$

where $U' = \{u'_1, u'_2, ..., u'_T\}$ are the system utterances sampled by $\mathcal{M}$ from prompts $X = \{x_1, x_2, ..., x_T\}$ at each turn.

Since the component to be trained is not the NLG itself but the GenPPN, our optimization is independent of the architecture and differentiability of NLG modules. In the following subsections, we describe the GenPPN, reward design for optimizing Eq. (2), and RL algorithm used.

### 3.2 Generative Post-processing with PLM

We assume that the model $\mathcal{M}$ of GenPPN is a language model based on the Transformer architecture (Vaswani et al., 2017). At turn $t$, the input prompt $x_t$ is created from the previous dialogue context $h_t$, DA $a_t$, and system utterances $u_t$ by using a hand-crafted prompt (specific prompts are shown in Section A.2 in the appendix).

Through RL, $\mathcal{M}$ learns to generate a rewritten system utterance $u'$ such that the dialogue result $e$ is maximized. However, even in the early stages of learning, if the system cannot produce at least reasonable system utterances, the dialogue always breaks down, and the training does not progress properly. For this reason, we adopt an instruction-tuned PLM (Wei et al., 2022; Chung et al., 2022; Taori et al., 2023), which has shown high performance on many NLP benchmarks including task-oriented dialogue modeling in few- and zero-shot settings (Pan et al., 2023; Hudeček and Dusek,

2023). Since the general PLM has a massive number of parameters, we use Low-Rank Adaptation (LoRA) (Hu et al., 2022), which has good parameter efficiency. That is, a small number of parameters $\theta$ is added to each self-attention module in the PLM, and only $\theta$ is optimized during training.

### 3.3 Reward with DA Contribution

Eq. (2) implies that only $e$ or a dialogue-level reward at the end of a multi-turn dialogue should be used to evaluate the generated utterances $U'$ for all $T$ turns. However, since this reward is too sparse and it is not feasible to optimize $\mathcal{M}$, we approximate Eq. (2) by using the utterance-level reward $r$, which evaluates how much of an effect each utterance has on the final task completion:

$$\theta^* = \arg\max_\theta \mathbb{E}_{U' \sim \mathcal{M}_\theta(X)} \left[ \sum_{u'_t \in U'}^T r(u'_t) \right] \quad (3)$$

The role of NLG is to accurately convey DAs to the user, i.e., to generate utterances such that $a_t = \hat{a}_t$, but this kind of evaluation based on the consistency of $a_t$ and $\hat{a}_t$ does not take into account $e$, so it cannot be used for $r$ directly.

With this in mind, we design $r$ by combining $e$ with the consistency evaluation for $a_t$ and $\hat{a}_t$. Specifically, we statistically measure the contribution of correctly conveying $(I, s, v) \in a_t$ to the user in terms of the impact on $e$ on the basis of the dialogues we have sampled. Then, we give a contribution-weighted reward to the utterance $u'_t$ that succeeds in conveying $(I, s, v)$.

To realize this, first, at the end of each dialogue, triples $\{(a_t, \hat{a}_t, e) | t \in T\}$ consisting of $a_t$ and $\hat{a}_t$ at each turn together with $e$ are added to the DA history $\mathcal{D}$. Here, $e$ is determined retrospectively on the basis of the outcome at the end of the dialogue; $e$ of all turns in a successful dialogue is 1, $\{(a_t, \hat{a}_t, 1) | t \in T\}$, and $e$ of all turns in a failed dialogue is 0, $\{(a_t, \hat{a}_t, 0) | t \in T\}$. Here, all $(a, \hat{a})$ pairs in $\mathcal{D}$ sampled so far are split into $\mathcal{S}$ and $\mathcal{F}$ depending on whether the task was successful:

$$\mathcal{S} = \{(a, \hat{a}) \mid (a, \hat{a}, e) \in D, e = 1\}$$
$$\mathcal{F} = \{(a, \hat{a}) \mid (a, \hat{a}, e) \in D, e = 0\}$$

Then, for each $(I, s, v) \in a_t$, the contribution $c_{(I,s)}$ of $(I, s)$ is calculated as follows:

$$c_{(I,s)} = \frac{n_{(I,s)}^{\text{Rec},\mathcal{S}} + n_{(I,s)}^{\text{Unr},\mathcal{F}}}{n_{(I,s)}^{\text{Rec},\mathcal{S}} + n_{(I,s)}^{\text{Rec},\mathcal{F}} + n_{(I,s)}^{\text{Unr},\mathcal{S}} + n_{(I,s)}^{\text{Unr},\mathcal{F}}} \quad (4)$$

where $n_{(I,s)}^{\text{Rec},\mathcal{S}}$ is the number of times that $(I, s)$ has been correctly recognized by the user (subscript "Rec" for "recognized") in a successful dialogue $\mathcal{S}$:

$$n_{(I,s)}^{\text{Rec},\mathcal{S}} = \sum_{(a,\hat{a}) \in \mathcal{S}} \sum_{(I',s',v') \in a \cap \hat{a}} [(I, s) = (I', s')]$$

Similarly, $n_{(I,s)}^{\text{Unr},\mathcal{S}}$ indicates the number of times in $\mathcal{S}$ that $(I, s)$ was not recognized by the user (subscript "Unr" for "unrecognized"), $n_{(I,s)}^{\text{Rec},\mathcal{F}}$ indicates the number of times in $\mathcal{F}$ that $(I, s)$ was recognized by the user, and $n_{(I,s)}^{\text{Unr},\mathcal{F}}$ indicates the number of times in $\mathcal{F}$ that $(I, s)$ was not recognized by the user.

Since $c_{(I,s)}$ is the co-occurrence probability of $(I, s)$ being recognized (or unrecognized) and the task being a success (or a failure), this quantifies how accurately conveying $(I, s)$ leads to task success. Also, $c_{(I,s)}$ is a value that is updated as the number of dialogue samples increases during training. That is, $c_{(I,s)}$ can be computed adaptively to the current performance of the GenPPN at each learning step. Note that $v$ is not taken into account for each count because the possible values of $v$ are so large (e.g., instances of phone number, address, etc.) that counting them separately would yield unreliable statistics.

Finally, the reward $r(u'_t)$ for the utterance generated by GenPPN at turn $t$ is calculated using this DA contribution $c_{(I,s)}$ as follows:

$$r(u'_t) = \text{Aggregate}(w(a_t, \hat{a}_t)) \quad (5)$$

$$w(a_t, \hat{a}_t) = \{\tau \cdot c_{(I,s)} \mid (I, s, v) \in a_t \cap \hat{a}_t\} \\ \cup \{-\tau \cdot c_{(I,s)} \mid (I, s, v) \in a_t \cap \bar{\hat{a}}_t\} \quad (6)$$

Here, the DAs correctly recognized by the user $(a_t \cap \hat{a}_t)$ are given a default score $\tau$, and conversely, the DAs not recognized by the user $(a \cap \bar{\hat{a}}_t)$ are given a negative score $-\tau$, and $w(a_t, \hat{a}_t)$ is the set of these scores weighted by each DA contribution $c_{(I,s)}$. The constant $\tau$ is a hyperparameter, and "Aggregate" is a function for aggregating $w(a_t, \hat{a}_t)$ into a scalar value for the reward. We design the following two types of Aggregate functions and empirically determine which one is better:

**mean** Output the mean of $w(a_t, \hat{a}_t)$

**absmax** Output the value with the highest absolute value among $w(a_t, \hat{a}_t)$. This is to emphasize the weighted score of $(I, s, v)$ with the highest contribution.

At the beginning of learning, the number of DAs recorded in $\mathcal{D}$ is small, so it is expected that an appropriate $c_{(I,s)}$ cannot be calculated. Therefore, prior to learning, $(a, \hat{a})$ is sampled and recorded in $\mathcal{D}$ by conducting multiple dialogues between the dialogue system and the user simulator without using GenPPN.

## 3.4 Optimization via RL

Following Stiennon et al. (2020), the RL algorithm used in this method is Proximal Policy Optimization (PPO) (Schulman et al., 2017). As a value network, a linear layer that outputs a scalar value randomly initialized with the parameter $\phi$ is added, and the overall trainable parameters are set to $\psi = [\theta; \phi]$. Also, following (Ziegler et al., 2019), to prevent the probability distribution of $\mathcal{M}_\psi$ from deviating too far from that of the original $\mathcal{M}$ due to parameter updates and thus losing its naturalness, a Kullback-Leibler (KL) divergence penalty is added to $r(u'_t)$ as the final reward $R_t$ for utterance $u'_t$ at turn $t$:

$$R_t = r(u'_t) - \beta \log \frac{\mathcal{M}_\psi(u'_t|x_t)}{\mathcal{M}(u'_t|x_t)} \qquad (7)$$

The clipped surrogate objective $\mathcal{L}(\psi)$ (Schulman et al., 2017) is used to optimize $\psi$ with the advantage estimated from the reward and value network. Since we need the user's subjective understanding results $\hat{a}$, this study uses a user simulator to optimize the GenPPN. The learning algorithm is summarized in Algorithm 1 in the appendix.

## 4 Experiments

In our experiments, we first evaluated the effectiveness of GenPPNs using a user simulator. Then, using the optimized GenPPN, we conducted a dialogue evaluation experiment using human subjects.

### 4.1 Dataset and Platform

We evaluated the effectiveness of our GenPPN using a dialogue system and a user simulator implemented on the basis of the MultiWOZ dataset (Budzianowski et al., 2018). MultiWOZ is a task-oriented dialogue dataset between a clerk and a customer at a travel information center, collected in Wizard-of-OZ style. It contains a variety of tasks across a total of seven domains (attraction, hotel, hospital, restaurant, taxi, train, and police).

We used ConvLab-2 (Zhu et al., 2020), a platform for evaluating task-oriented dialogue systems

that provide various modules for dialogue systems, a user simulator, and an evaluation tool. The following describes the dialogue system, user simulator, and user goals used in this experiment.

**Dialogue System** To make it easier to assess changes in the performance of utterance generation, the other modules (that is, NLU, DST, and Policy) should have stable performance. Therefore, we used the best-performing BERT (Devlin et al., 2019)-based NLU (Chen et al., 2019), rule-based DST, and rule-based Policy available in ConvLab-2. BERT-based NLU is a model that uses representations embedded by BERT to classify user intentions in user utterances and extract slots by sequence labeling. Rule-based DST and Policy are modules implemented using hand-crafted rules. Three models were selected as NLG modules (see Section 4.2 for the NLG models used), and GenPPN was applied to each of them to verify its generality.

**User Simulator** For the user simulator, we used a combination of BERT-based NLU, agenda-based Policy (Schatzmann et al., 2007), and template-based NLG. The agenda-based Policy models a user's behavior in MultiWOZ by using a stack-like agenda created using hand-crafted rules.

**User Goal** A user goal for each dialogue is randomly generated; the domains are randomly selected from one to three domains (out of all seven domains). The slots are also randomly selected on the basis of the slots' frequency in MultiWOZ.

### 4.2 NLG Baselines

We applied GenPPN to each of the three NLGs available in ConvLab-2 with different architectures in order to demonstrate that it works for a variety of NLGs. In addition, one NLG optimized with only utterance-level rewards, without considering task success, was also evaluated for comparison.

**Template NLG** An NLG model that uses the template utterances representing each DA. Because each utterance is carefully designed by hand, this model has significantly higher performance than other NLG baselines (Takanobu et al., 2020).

**SC-LSTM (Wen et al., 2015)** An LSTM-based model with a reading gate mechanism. This model takes binary feature vectors representing DAs as context and decodes utterances.

| NLG | Task Success | Inform | | | Book Rate | Turn ↓ | DA F1 |
| --- | --- | --- | --- | --- | --- | --- | --- |
| | | F1 | Precision | Recall | | | |
| Template NLG | 77.25 | 78.44 | 74.19 | 89.13 | 83.91 | 7.67 | 71.73 |
| + GenPPN$_{mean}$ | 77.93 | 79.75 | **75.62** | 89.41 | 84.33 | 7.63 | 76.98 |
| + GenPPN$_{absmax}$ | **78.91**$^*$ | **79.86** | 75.58 | **89.93** | **85.19** | **7.02** | **78.23** |
| SC-LSTM | 54.00 | 67.45 | 68.07 | 72.48 | 67.69 | 11.65 | 60.56 |
| + GenPPN$_{mean}$ | 60.64$^{**}$ | 75.40 | 75.42 | 81.01 | **78.74** | 9.42 | **79.80** |
| + GenPPN$_{absmax}$ | **72.95**$^{**}$ | **79.46** | **77.38** | **86.16** | 78.46 | **7.21** | 79.08 |
| SC-GPT | 64.94 | **78.06** | **73.60** | 88.51 | 56.94 | 7.80 | 71.53 |
| + GenPPN$_{mean}$ | **73.63**$^{**}$ | 76.54 | 71.81 | 87.79 | **82.08** | 8.03 | 73.07 |
| + GenPPN$_{absmax}$ | 73.34$^{**}$ | 77.34 | 72.29 | **89.03** | 80.79 | **7.50** | **73.72** |
| GPT-2 + RL | 72.36 | 76.70 | 73.50 | 85.99 | 76.81 | 7.47 | **81.17** |
| + GenPPN$_{mean}$ | 74.02 | 77.10 | 73.87 | 86.82 | 79.19 | 7.54 | 80.98 |
| + GenPPN$_{absmax}$ | **75.20**$^{**}$ | **78.79** | **75.58** | **88.02** | **79.80** | **7.15** | 80.08 |

Table 1: Performance of dialogue system with each NLG and with GenPPN applied to them. Subscripts in GenPPN indicate Aggregate function used in reward calculation. Highest score for each NLG is in bold. $*$ and $**$ indicate Task Success for GenPPN and were significantly better than original NLGs at $p < 0.05$ and $< 0.01$, respectively, in McNemar test with Bonferroni correction.

**SC-GPT ([Peng et al., 2020](#))** A GPT-2 based model that generates utterances from DA text sequences. This has been trained on seven task-oriented dialogue corpora including MultiWOZ and the Schema-Guided Dialogue Dataset ([Rastogi et al., 2020](#)), and it is a SOTA on the MultiWOZ NLG benchmark.

**GPT-2 + RL ([Ohashi and Higashinaka, 2022a](#))** A GPT-2-based NLG model optimized with an utterance-level reward. This model was trained to maximize the accuracy of $a$ and $\hat{a}$ (as measured by F1) using the DA-system utterance pairs in the MultiWOZ corpus in an offline fashion.

Note that, since the purpose of our experiment is to verify whether GenPPN can enhance the performance of NLG irrespective of its architecture or base performance, the verification of methods that fine-tune NLG models themselves is outside the scope of this study.

### 4.3 Evaluation Metrics

In the evaluation of the dialogue system, we used common metrics for task-oriented dialogues: *Turn, Inform F1/Precision/Recall, Book Rate, Task Success*. Turn indicates the number of turns required for each dialogue; the smaller it is, the more efficiently the dialogue can be conducted. Inform F1/Precision/Recall indicate whether the system responded appropriately to the user's requests. Book Rate indicates whether the entity booked by the system correctly matched that of the user's goal.

Task Success is evaluated by calculating whether Inform Recall and Match Rate both became one within the maximum number of turns. To evaluate the utterance-level performance of NLG, we additionally used *DA F1*, which measures the match rate of $a$ and $\hat{a}$ at each turn by F1 ([Ohashi and Higashinaka, 2022a](#); [Guo et al., 2023](#)).

### 4.4 Training GenPPN

Stanford Alpaca 7B ([Taori et al., 2023](#)) was used as the instruction-tuned PLM for the GenPPN. This model is a fine-tuned version of LLaMA-7B ([Touvron et al., 2023](#)) with a 52K instruction dataset. The input prompts include the dialogue history, the previous DA, the system utterances generated by NLG from that DA, and the instructions for rewriting that system utterance (specific prompts are shown in Section A.2 in the appendix). Throughout the experiment, we first sampled 1K dialogues to initialize the DA history and then trained 200 iterations. See Section A.3 for further training details.

For the evaluation, the GenPPN at the highest-reward iteration was used, and dialogue simulations were performed using 1,024 user goals prepared specifically for the test. We report the average scores in this paper.

### 4.5 Main Results

Table 1 shows the evaluation results[2]. For Template and SC-LSTM, GenPPN improved all evalua-

---

[2]We used the latest version of ConvLab-2 as of June 2023; we could not reproduce the scores reported on the leaderboard for some baseline systems. The main reason is probably due to the difference in the random seed and BERT NLU weights.

| NLG | Success | Inf. F1 | Book | DA F1 |
|---|---|---|---|---|
| SC-LSTM + GenPPN | **72.95** | **79.46** | 78.46 | 79.08 |
| w/o adaptive $c_{(I,s)}$ | 62.11 | 74.80 | 78.15 | 70.12 |
| w/o $e$ | 59.47 | 77.02 | 71.06 | 76.00 |
| w/o $r(u')$ | 63.67 | 77.58 | **80.83** | 62.69 |
| w/o context | 68.46 | 76.25 | 74.17 | **80.28** |
| w/o response | 47.36 | 68.96 | 74.67 | 65.21 |

Table 2: Ablation results when applying GenPPN to SC-LSTM. "absmax" Aggregate function is used in calculation of rewards using DA contributions.

| NLG | $N$ | Success | Turns | Und. | App. | Sat. |
|---|---|---|---|---|---|---|
| SC-LSTM | 54 | 33.33 | 17.72 | **4.09** | **4.17** | **4.04** |
| + GenPPN | 50 | **52.00**[*] | **15.08**[**] | 4.04 | 4.14 | 3.98 |

Table 3: Human evaluation results. $N$ is number of subjects who interacted with each system. "Und.," "App.," and "Sat." represent user's subjective evaluation of system's understanding, appropriateness of system utterance, and satisfaction with dialogue, respectively. Scores of SC-LSTM and our GenPPN were compared using the McNemar test; significant differences are indicated with $*$ ($p < 0.05$) and $**$ ($p < 0.01$).

tion metrics. The absmax aggregation function improved the final performance more than the mean. For SC-LSTM, Task Success was nearly 19 points better than the original SC-LSTM, especially when using absmax. These results indicate that a better strategy for evaluating each utterance is to prioritize learning to convey the DA with the highest contribution rather than to take the average. Considering that the improvement in Template NLG was only about 1.7%, the lower the performance of the original NLG, the more room there is for improvement by GenPPN.

For SC-GPT, Task Success, Book Rate, and Turn steadily improved. However, Inform Precision slightly decreased, unlike other cases where GenPPN was applied to other NLG modules. This suggests that GenPPN learned that improving Book Rate is more important for improving Task Success, even at the expense of Inform Precision.

GPT-2 + RL has a higher DA F1 than any of the GenPPNs. This is reasonable, given that GPT-2 + RL is optimized using DA F1 as a reward. GenPPN was able to further improve GPT-2 + RL's Task Success while the utterance-level metric DA F1 slightly decreased from the original score. This suggests that not only utterance-level rewards but dialogue-level rewards must be introduced to improve the task completion performance of the entire system.

Note that the performance of SC-LSTM + GenPPN and SC-GPT + GenPPN is lower than that of Template NLG baseline. However, the primary goal of our study is not to obtain an NLG model with SOTA performance but rather to enhance the performance of NLG irrespective of its architecture or its base performance. In our experiments, the dialogue performance of NLG models like SC-LSTM and SC-GPT, in addition to Template NLG, were improved using GenPPN, showing that our primary goal has been achieved.

## 4.6 Ablation Study

We analyzed the impact of each factor in the GenPPN optimization on the final performance. For the NLG model in this ablation study, we used SC-LSTM, which showed the greatest improvement in performance over the other NLGs in Table 1. Table 2 shows the results.

**Adaptive DA Contribution** "w/o adaptive $c_{(I,s)}$" is a reward design that does not update the DA history, and in all steps, uses the DA contribution calculated only from the 1,000 dialogues sampled prior to training. This resulted in a 10 points decrease in Task Success. Therefore, we found that it is important to constantly update the DA contributions as the GenPPN changes.

**Dialogue-level Reward** "w/o $e$" does not use DA contribution, and only an utterance-level reward based on the accuracy (F1) between $a$ and $\hat{a}$ is used. That is, the reward design does not take into account the dialogue-level reward $e$ related to task completion. As a result, Task Success decreased significantly, while DA F1 did not decrease much. Therefore, it was shown that the DA contribution is a useful factor in improving the task success rate.

**Utterance-level Reward** "w/o $r(u')$" does not use the utterance-level reward $r(u')$ but only the last dialogue evaluation result as a reward. That is, all utterances were given the dialogue result $e \in \{0, 1\}$ equally. Although Book Rate improved, Inform F1 did not, indicating that it is difficult to optimize GenPPN with only a sparse $e$ reward, as described in Section 3.3.

**Dialogue Context** "w/o context" indicates that the input of GenPPN does not include the dialogue history. The slight decrease in Task Success, Inform F1, and Book Rate indicates that GenPPN should make use of the previous history to improve

| Context | **User:** Hello, can you help me find a restaurant for my upcoming trip to Cambridge? Hmm, I 'll try Asian oriental food. I'm looking for an expensive restaurant. I also would like information on a place to eat in the centre. **System:** kymmoy serves asian oriental food. **User:** May I have the address? I would like their phone number. |
|---|---|
| System DA | {(Inform, Attraction-Addr, pool way, whitehill road), (Inform, Attraction-Phone, 01223902088), (Inform, Restaurant-Addr, 52 Mill Road City Centre), (Inform, Restaurant-Phone, 01223311911)} |
| System Response | **SC-LSTM:** Their address is / **+ GenPPN:** Their address is 52 Mill Road City Centre and phone number is 01223311911. |
| DA predicted by user | {} / {(Inform, Restaurant-Addr, 52 Mill Road City Centre), (Inform, Restaurant-Phone, 01223311911)} |

Table 4: Response examples of systems using SC-LSTM and SC-LSTM + GenPPN in same context.

dialogue performance. Meanwhile, the improvement in DA F1 suggests that dialogue history is not necessary if we only want to optimize at the utterance level.

**System Response** "w/o response" does not rewrite the system utterance, but GenPPN generates it directly from the dialogue history and DA. It can be seen that the performance of all the methods significantly degraded. The reason for this is that the untrained GenPPN, i.e., Alpaca-7B, has never seen the mapping from DA to system utterance in MultiWOZ and thus has difficulty generating system utterances in a zero-shot setting. Therefore, it is considered more reasonable to learn to rewrite utterances rather than to generate them from scratch.

### 4.7 Human Evaluation

We tested whether GenPPN optimized by dialogue simulation is also effective for humans. In this experiment, we also used the system using SC-LSTM, which showed the best performance improvement over the other NLGs in Table 1. Over 50 crowd workers were recruited via Amazon Mechanical Turk, and each worker interacted once with one of the systems using SC-LSTM or SC-LSTM + GenPPN for up to 20 turns after being instructed about their user goal. Each worker had 20 turns to judge whether or not the task was successful; after 20 turns, the task was forced to fail. After the dialogue was completed, each worker was asked to rate the system's ability to understand the language (Und.), the accuracy of the system's responses (App.), and overall satisfaction with the dialogue (Sat.) on a 5-point Likert scale. See Section A.5 in the appendix for more details on the human evaluation settings.

Table 3 shows the results of the evaluations of each system. GenPPN significantly improved in terms of both Task Success and Turn, which are

measures of task completion. However, the user's subjective ratings of understanding, appropriateness, and dialogue satisfaction showed no improvement, which is reasonable given that GenPPN is optimized for task completion only. This also means that GenPPN did not generate unnatural utterances to improve Task Success and Turn, which is considered a positive result. We would like to evaluate the effectiveness of GenPPN for other NLG baselines than SC-LSTM by humans in the future.

## 5 Case Study

Table 4 compares the behavior of two systems using SC-LSTM and using SC-LSTM + GenPPN in a dialogue. As a context, the user requested the address and phone number of the restaurant, but the Policy module of the dialogue system mistakenly decided to answer with the address and phone number of an attraction as well. Here, SC-LSTM was not able to generate sentences because there was no example in its training data, in which DAs in different domains occur at the same time. As a result, SC-LSTM failed to convey any information to the user. In contrast, GenPPN reflected the address and phone number of the restaurant in the post-processed utterance, and as a result, the information was correctly conveyed as requested by the user, and the task was successful.

It is worth noting that the utterance generated by post-processing does not contain any information about the attraction. This is probably because GenPPN judged from the dialogue context that the information about the attraction was not what the user requested, and that conveying this piece of information would interfere with the task completion. These results indicate that GenPPN can correct errors propagated from preceding modules beyond the realm of NLG so that the entire system's performance improves.

## 6 Summary and Future Work

Post-Processing Networks (PPNs), which modify the outputs of arbitrary modules including non-differentiable ones in task-oriented dialogue systems, have been applied to NLU, DST, and Policy modules. In this paper, we proposed Generative Post-processing Networks (GenPPNs) for NLG modules. We optimized GenPPN toward the task completion performance of the entire system by using DA contribution-based rewards. Experiments on the MultiWOZ dataset and dialogue simulations confirmed that our GenPPN could improve the task success rate of various dialogue systems, regardless of the NLG architecture. It was also confirmed that a GenPPN optimized with a dialogue simulation was effective in a human evaluation experiment.

In the future, we would like to integrate our GenPPN with the existing PPNs of NLU, DST, and Policy to realize the post-processing of all modules and further improve the systems performance. Since GenPPN is a generative model, we would like to extend it not only to text but also to the post-processing of speech recognition and speech synthesis modules in spoken dialogue systems.

## Limitations

Our work has several limitations, which we aim to address in our future work.

First, this study implemented, trained, and evaluated the GenPPN using only the MultiWOZ dataset. Therefore, the applicability of the GenPPN to task-oriented dialogues other than MultiWOZ has not been verified. In particular, the DA-dependent reward design in this study needs to be improved for application to dialogues such as open-domain chats, where ontologies such as intent, slot, and value are not defined.

Second, this study used a user simulator for training the GenPPN. When applied to a new domain, the cost of implementing a user simulator is likely to be high. Therefore, a user simulator-free learning method is needed in this respect. In addition, a GenPPN optimized by a user simulator learns user-simulator-specific utterances, which may not be the best utterances for humans. To learn the optimal GenPPN for humans, it is essential to design a reward model that takes humans into account, using a human-in-the-loop approach.

Third, this study required running a PLM with a relatively large parameter size. During inference, in addition to the system's response generation, the GenPPN is required to rewrite utterances, which is expected to increase the operation cost and generation time. In addition, more computational resources are required during learning. Therefore, it is necessary to study ways to reduce learning and inference costs.

## Ethics Statement

We used the publicly available MultiWOZ dataset in full compliance with its terms of use. Our use of the LLaMA language model was also entirely consistent with the prescribed usage guidelines. No private or confidential data was used at any stage of the research. For our human evaluation experiment, we underwent and passed an ethical review from our institution and strictly adhered to its rules and guidelines. The anonymity, privacy, and rights of the subjects were preserved throughout. We acknowledge that there are potential ethical concerns associated with the use of large pre-trained language models, such as the risk of generating harmful or discriminatory statements.

## Acknowledgments

This work was supported by JSPS KAKENHI Grant Number 19H05692. We used the computational resources of the supercomputer "Flow" at the Information Technology Center, Nagoya University.

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

# A Appendix

## A.1 GenPPN Optimization Algorithm

---

**Algorithm 1** Optimization of GenPPN via PPO

---

**Require:** Dialogue system $\mathcal{A}$, User simulator $\mathcal{U}$
**Require:** GenPPN $\mathcal{M}$
1: Initialize DA history $\mathcal{D}$ by sampling several dialogues using $\mathcal{A}$ and $\mathcal{U}$
2: Prepare $\mathcal{M}_{\psi_{\text{old}}}$ with randomly initialized LoRA parameters $\theta$ and value network parameters $\phi$
3: **for** each training iteration **do**
4:     **while** #turns does not reach batch size **do**
5:         Sample a dialogue and $(a_t, \hat{a}_t)$ for each turn $t$ using $\mathcal{A}$, $\mathcal{M}_{\psi_{\text{old}}}$, and $\mathcal{U}$
6:         Obtain final evaluation result $e$
7:         Add $(a_t, \hat{a}_t, e)$ of each $t$ to $\mathcal{D}$
8:     **end while**
9:     Calculate reward $R_t$ using Eq. (7)
10:     Compute advantage estimates
11:     Optimize $\mathcal{L}(\psi)$ with a certain epoch and mini-batch size
12:     Update $\psi_{\text{old}} \leftarrow \psi$
13: **end for**

---

## A.2 Prompt for GenPPN

At a given turn $t$, the input prompts to GenPPN include the DA $a_t$ output by the system's Policy module and the original system utterance $u_t$ generated by the NLG from $a_t$. As the dialogue context, we also included user utterances at $t-1$, DAs $a_{t-1}$, system utterances $u'_{t-1}$ generated by GenPPN, and user utterances at $t$. The format of the instructional text in the prompts was adapted from the phrases in the Alpaca dataset (Taori et al., 2023). Examples of prompts are shown in Figure 3. In our experiments, due to the maximum input length of the model, the dialogue history was limited to the past one turn.

## A.3 Details of GenPPN Training

As the model for GenPPN, we used Stanford Alpaca-7B trained weights for the prompts, which are publicly available on HuggingFace Hub[3]. The maximum number of input tokens for a prompt was set to 512, and the maximum number of generated tokens was set to 128. When generating utterances, sampling was performed with the beam size set to 1, temperature to 1.0, and top-$p$ to 1.0. The $\tau$ used in the reward calculation (Eq. (6)) was set to 1.

---

[3] https://huggingface.co/tatsu-lab/alpaca-7b-wdiff

| Hyperparameter Name | | Value |
|---|---|---|
| LoRA | Target projection matrix of self-attention module | query, key value, output |
| | Rank | 16 |
| | Scaling factor $\alpha$ | 16 |
| PPO | Total iterations | 200 |
| | Total batch size | 512 |
| | Epoch | 4 |
| | Total mini-batch size | 32 |
| | Learning rate | 1e-5 |
| | Optimizer | Adam |
| | Discount factor $\gamma$ | 1.0 |
| | GAE factor $\lambda$ | 0.95 |
| | Clipping $\epsilon$ | 0.2 |
| | Coef. of KL penalty $\beta$ | 0.01 |

Table 5: Hyperparameter settings

Table 5 shows the hyperparameters of LoRA and PPO during training. Throughout the experiment, we first sampled 1,000 dialogues to initialize the DA history and then trained 200 iterations, with a batch size of 512 turns per iteration (corresponding to about 50 dialogues) and a fixed learning rate of 1e-5 with Adam Optimizer (Kingma and Ba, 2014). We used $16 \times$ V100 32-GB computational resources. Training of 200 iterations took about seven hours to complete.

Figure 4 shows the training curves when GenPPN is applied to Template NLG, SC-LSTM, SC-GPT, and GPT-2 + RL respectively. The figure shows that both the reward and Task Success are low at the start of training (i.e., before RL is applied) and improve as RL progresses, which means that fine-tuning LLMs with RL is important for our GenPPN. We have yet to investigate the effects of instruction tuning. We plan to validate this by comparing the performance of instruction-tuned models other than Alpaca (e.g., Flan-T5 (Chung et al., 2022)) and non-instruction-tuned models (e.g., LLaMA (Touvron et al., 2023)).

## A.4 Additional Ablations

In addition to the study on SC-LSTM in Section 4.6, we conducted the same ablation study on two higher-performing NLGs: Template NLG and GPT-2 + RL. Table 6 shows the results. In Template NLG, the improvement with GenPPN was the greatest when no context was used. Nevertheless, for both Template NLG and GPT-2 w/ RL, there was a trend showing that adaptive DA contribution was important for performance improvement.

> Below is an instruction that describes a task. Write a response that appropriately completes the request.
>
> ### Instruction:
> Begin by reading a conversation between a customer and a chatbot about travel information.
>
> Customer: I am looking for a hotel call kirkwood house . I need a hotel as well .
> Chatbot Action: Inform-Hotel(Price=moderate; Area=north; Type=guesthouse; Parking=yes; Stars=4; Name=kirkwood house; Internet=yes)
> Chatbot: kirkwood house is a moderate 4 star guesthouse in the north.
> Customer: Do they have free parking ?
> Chatbot Action: Inform-Hotel(Name=kirkwood house; Parking=yes)
> Chatbot: kirkwood house is available.
>
> Your task is to rephrase the chatbot's last utterance so the customer can understand it. Make sure to include the content of \`Chatbot Action:\` in the utterance. If no rephrasing is necessary, repeat the original utterance.
>
> ### Response:
> Chatbot:

Figure 3: Examples of prompts entered into GenPPN. "Customer:", "Chatbot Action:", and "Chatbot:" denote user utterance, system DA, and system utterance at each turn, respectively.

| NLG | Success | Inf. F1 | Book | DA F1 | NLG | Success | Inf. F1 | Book | DA F1 |
|---|---|---|---|---|---|---|---|---|---|
| Template NLG + GenPPN | 78.91 | 79.86 | 85.19 | 78.23 | GPT-2 + RL + GenPPN | **75.20** | **78.79** | 79.80 | 80.08 |
| w/o adaptive $c_{(I,s)}$ | 76.27 | 78.07 | 83.00 | 75.05 | w/o adaptive $c_{(I,s)}$ | 67.68 | 76.75 | 79.31 | 81.04 |
| w/o $e$ | 77.15 | 79.29 | 83.30 | 76.89 | w/o $e$ | 72.56 | 76.57 | 77.36 | 80.63 |
| w/o $r(u')$ | 71.09 | 78.14 | 80.89 | 63.68 | w/o $r(u')$ | 72.85 | 78.52 | **81.82** | 71.15 |
| w/o context | **79.00** | **79.92** | **85.29** | **78.58** | w/o context | 73.24 | 77.18 | 77.14 | **85.11** |

(a) Template NLG

(b) GPT-2 + RL

Table 6: Ablation results when applying GenPPN to Template NLG and GPT-2 + RL. "absmax" Aggregate function is used in calculation of rewards using DA contributions. **Bold** and underlined scores indicate top and second top, respectively.

## A.5 Details of Human Evaluation

In collecting participants for the human evaluation via Amazon Mechanical Turk (AMT), we recruited workers who met the following four conditions: (1) had residence in an English-speaking county, (2) had at least 100 completed Human-Intelligence Tasks (HIT) on AMT, (3) had an acceptance rate of 95% or higher on the HIT on AMT, and (4) answered all five common sense questions correctly.

In the experimental procedure, each worker was first instructed about a certain user goal. The user goal was randomly generated by a program as in the dialogue simulation. Considering the high comprehension ability of humans, the number of domains included in the user goal was set to 3, which is a higher difficulty level than in the simulation. After reading the user goal, the workers interacted with either the system using SC-LSTM or the system using SC-LSTM + GenPPN. The maximum number of turns in the dialogue was 20, and each worker judged whether the task was completed within the

maximum number of turns. After 20 turns, the task was forced to fail. After the dialogue, the workers answered the three-question questionnaire described in Section 4.7.

Considering that each HIT should take about 10 minutes to complete, the reward was set at $2, and the time limit was 30 minutes. The number of subjects was not equal because the system was randomly selected each time, resulting in 54 evaluators for SC-LSTM and 50 evaluators for SC-LSTM + GenPPN.

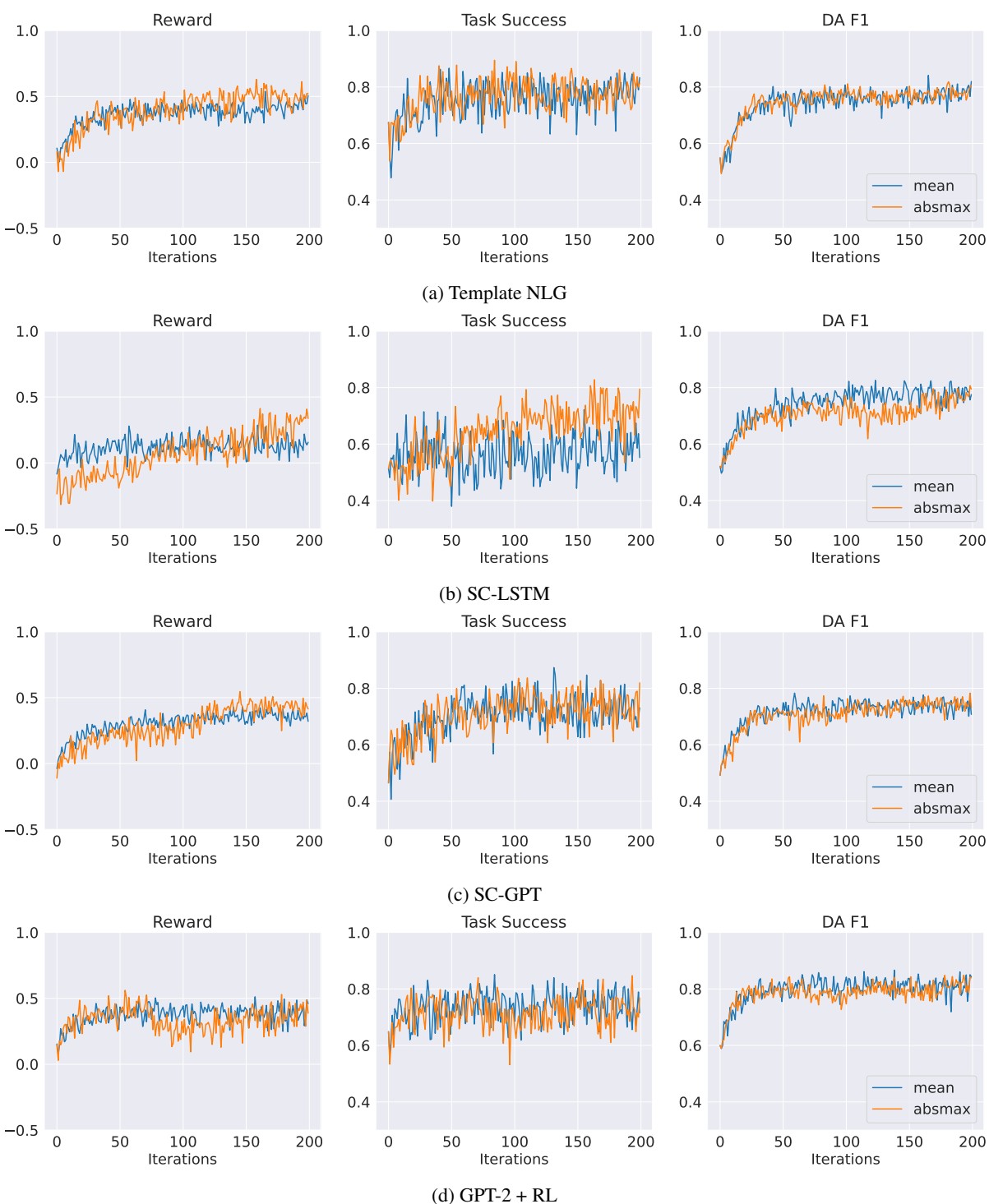

Figure 4: Training curves when GenPPN is applied to each NLG. Either "mean" or "absmax" is used as aggregation function. "Reward" indicates value calculated by Eq. (5).