# OpenReview forum: "Enhancing Task-oriented Dialogue Systems with Generative Post-processing Networks"
_EMNLP/2023/Conference — EMNLP 2023 Main_

### Official Review · Reviewer_zF36 · 2023-08-05

**Soundness:** 4

**Excitement:**

3: Ambivalent: It has merits (e.g., it reports state-of-the-art results, the idea is nice), but there are key weaknesses (e.g., it describes incremental work), and it can significantly benefit from another round of revision. However, I won't object to accepting it if my co-reviewers champion it.

**Missing References:**

1. Adaptive Natural Language Generation for Task-oriented Dialogue via Reinforcement Learning

**Paper Topic And Main Contributions:**

This paper proposes a generative post-processing network to modify the output of NLG component in the pipeline task-oriented dialogue (TOD) system. The model was built using LLaMA language model and optimized using PPO with the predefined reward function.

**Questions For The Authors:**

1. How did you run automatic evaluation in ConvLab-2? I wonder why the result of template-based NLG did not reach more than 82% of success rate? did you follow standard ConvLab-2 evaluation as mentioned in the DSTC-9 Track 2?
2. Why did you consider only SC-LSTM for ablation study and human evaluation?

**Reasons To Accept:**

- The idea of proposing post-processing network to modify the output of the TOD system is useful in practical.
- The experimental result shows that the proposed method can improve the performance of the TOD system

**Reasons To Reject:**

- The motivation is still limited, although it is practical solution. As an example, the reward definition is mainly designed based on the success rate. In fact, using template-based NLG under this motivation is enough in my opinion. If the author considers other criteria for the main motivation such as fluency, naturalness of the generated sentences then it will be more meaningful.
- The evaluations were highly intended to compare with SC-LSTM which is quite old method. If the evaluation is compared to the SOTA method such as "Adaptive Natural Language Generation for Task-oriented Dialogue via Reinforcement Learning" or template-based NLG then it will be better

**Reproducibility:**

4: Could mostly reproduce the results, but there may be some variation because of sample variance or minor variations in their interpretation of the protocol or method.

**Reviewer Confidence:**

5: Positive that my evaluation is correct. I read the paper very carefully and I am very familiar with related work.

---

> ### Author Rebuttal · Authors · 2023-08-29
>
> We are grateful to the reviewer for the thoughtful and encouraging review. We provide explanations and the new results of additional experiments to respond to your concerns and comments as follows.
>
> #### **Q1. The reward definition of reinforcement learning is mainly based on Task Success, but template-based NLG may be sufficient. Placing objectives on other criteria, such as fluency and naturalness, would be more meaningful.**
>
> The primary goal of our study is not to obtain an NLG model with state-of-the-art (SOTA) performance, but rather to enhance the performance of NLG irrespective of its architecture or its base performance. In our experiments, the dialogue performance of NLG models like SC-LSTM and SC-GPT, in addition to Template NLG, were improved using GenPPN, showing that our primary goal has been achieved.
>
> Although NLG baselines in our experiments included Template NLG, known for its high performance [1], in real-world situations, meticulously designed and high-performance NLG models like the template NLG are only sometimes available in other tasks or domains. We believe that GenPPN is a practical approach to improving NLGs in such scenarios.
>
> We also believe there is value in GenPPN when considering objectives other than task completion, such as enhancing naturalness and diversity. As additional experiments, we measured distinct-N of utterances generated by each NLG model during the test. The results are presented in the table below:
>
> | NLG | Success | Distinct-1 | Distinct-2 |
> | --- | :---: | :---: | :---: |
> | Template | 77.25 | 1.91 | 5.37 |
> | Template w/ GenPPN | 78.91 | 1.99 | 5.97 |
> | SC-LSTM | 54.00 | 3.38 | 5.46 |
> | SC-LSTM w/ GenPPN | 72.95 | 3.10 | 7.76 |
> | SC-GPT | 64.94 | 4.39 | 15.89 |
> | SC-GPT w/ GenPPN | 73.63 | 4.62 | 16.08 |
>
> The distinct-N of Template NLG is low, which indicates fixed and mechanical utterances. In contrast, when applying GenPPN to SC-LSTM or SC-GPT, we were able to improve the Task Success score to a level comparable to Template NLG while maintaining the original high distinct-N of the model. This result suggests that we achieved dialogue performance comparable to Template NLG with more diverse utterances.
>
> #### **Q2. In the evaluation, the focus is on comparing with the older SC-LSTM. It would be better if there were comparisons with SOTA methods like "Adaptive Natural Language Generation for Task-oriented Dialogue via Reinforcement Learning" [1] or template-based NLG.**
>
> As you pointed out, a study with the SOTA method would be better. As additional experiments, we first applied GenPPN to GPT-2 w/ RL [1] to test its effectiveness. The results are shown in the following table:
>
> | NLG | Success | Inform F1 | Inform Precision | Inform Recall | Book Rate | Turn | DA F1 |
> | --- | :---: | :---: | :---: | :---: | :---: | :---: | :---: |
> | GPT-2 w/ RL | 72.36 | 76.70 | 73.50 | 85.99 | 76.81 | 7.47 | **81.17** |
> | w/ GenPPN$_\text{mean}$ | 74.02 | 77.10 | 73.87 | 86.82 | 79.19 | 7.54 | 80.98 |
> | w/ GenPPN$_\text{absmax}$ | **75.20** | **78.79** | **75.58** | **88.02** | **79.80** | **7.15** | 80.08 |
>
> From the results, we observed that even for an NLG model optimized at the utterance-level, GenPPN was able to further improve its dialogue performance. However, the utterance-level metric DA F1 slightly decreased from the original GPT-2 w/ RL score, suggesting a limited correlation between dialogue-level and utterance-level performance.
>
> We also conducted ablation studies on SOTA models, GPT-2 w/ RL and Template NLG, and the results are as follows (**bold** and _italic_ values indicate the best and next best scores, respectively):
>
> | NLG | Success | Inform F1 | Book Rate | DA F1 |
> | --- | :---: | :---: | :---: | :---: |
> | GPT-2 w/ RL w/ GenPPN | **75.20** | **78.79** | _79.80_ | 80.08 |
> | w/o adaptive $c_{(I,s)}$ | 67.68 | 76.75 | 79.31 | _81.04_ |
> | w/o $e$ | 72.56 | 76.57 | 77.36 | 80.63 |
> | w/o $r(a, \hat{a})$ | 72.85 | _78.52_ | **81.82** | 71.15 |
> | w/o context | _73.24_ | 77.18 | 77.14 | **85.11** |
>
> | NLG | Success | Inform F1 | Book Rate | DA F1 |
> | --- | :---: | :---: | :---: | :---: |
> | Template w/ GenPPN | _78.91_ | _79.86_ | _85.19_ | _78.23_ |
> | w/o adaptive $c_{(I,s)}$ | 76.27 | 78.07 | 83.00 | 75.05 |
> | w/o $e$ | 77.15 | 79.29 | 83.30 | 76.89 |
> | w/o $r(a, \hat{a})$ | 71.09 | 78.14 | 80.89 | 63.68 |
> | w/o context | **79.00** | **79.92** | **85.29** | **78.58** |
>
> For GPT-2 w/ RL, trends similar to those of SC-LSTM were observed. Thus, it was evident that an adaptive DA contribution that considers both dialogue- and utterance-levels is crucial for Task Success.
>
> #### **Q3. Why does the performance of Template NLG not reach the 82% reported in the DSTC-9 Track 2 ConvLab-2 evaluation?**
> The difference in Task Success between the template NLG presented on the ConvLab-2 leaderboard (which we will refer to as ConvLab-2's baseline) at 81.3% and our baseline at 77.3% can be mainly attributed to (1) the setting of the NLU module of the user simulator and (2) the setting of the template NLG.
> 1. The NLU module of the user simulator in ConvLab-2’s baseline used the utterances in the last three turns as an additional context, while the NLU in our baseline did not. This setting without context is consistent with what was reported for the training of GPT-2 w/ RL [1], and we used this setting to ensure fair comparison in Section 4.5’s main result. This difference in configuration may have affected the performance.
> 2. For the Template NLG, there are two modes: the “manual mode” that uses only hand-crafted utterances and the “retrieval mode” that constructs utterances by retrieving actual utterances from the MultiWOZ corpus. In our experiments, we employed the basic manual mode, whereas ConvLab-2’s baseline may be using the retrieval mode.
>
> #### **Q4. Why were only SC-LSTM considered for ablation study and human evaluation?**
> As described in Sections 4.6 and 4.7, the improvement in Task Success of SC-LSTM w/ GenPPN is the largest compared to other NLG models, so we considered that using SC-LSTM is appropriate to analyze the impact of GenPPN.
>
> #### **References:**
> [1] Ohashi and Higashinaka, 2022. Adaptive Natural Language Generation for Task-Oriented Dialogue via Reinforcement Learning. In Proc. COLING2022, pages 242–252.

---

### Official Review · Reviewer_SfSx · 2023-08-07

**Soundness:** 4

**Excitement:**

3: Ambivalent: It has merits (e.g., it reports state-of-the-art results, the idea is nice), but there are key weaknesses (e.g., it describes incremental work), and it can significantly benefit from another round of revision. However, I won't object to accepting it if my co-reviewers champion it.

**Paper Topic And Main Contributions:**

The paper introduces a novel component called Generative Post-Processing Networks (GenPPNs) designed specifically for enhancing the quality of utterances generated by natural language generation (NLG) modules within task-oriented dialogue systems. GenPPNs leverage the previous dialogue context, dialogue action, and system utterance as input to generate a refined system output that is rewritten based on the original system utterance. To enhance the performance of GenPPNs using reinforcement learning, the paper proposes the incorporation of a dialogue act (DA) contribution, which facilitates the distribution of dialogue-level rewards to utterance-level rewards. This approach optimizes the learning process and improves the task completion performance of three distinct NLG modules, as evaluated on the MWOZ dataset. Moreover, the introduction of the DA contribution proves to be effective in facilitating the learning process of GenPPNs and further improving the overall task completion performance.


**Questions For The Authors:**

Question A: In the context of reward using DA contribution, whether the value of "e" can only be equal to 1 at the end of a dialogue? If this were the case, it raises the possibility of an imbalance between the Successful set and Failed set.
Question B: Why was the main experiment, involving GPT-2 with RL, not conducted using GenPPNs?
Question C: To what extent does the incorporation of task completion metrics in GenPPNs for RL optimization result in an excessive alignment of the final optimized model with the targeted metrics? More specifically, in the context of human evaluation experiments, why did the performance of GenPPNs decline across all three user subjectivity assessment metrics, and can this be attributed to the RL optimization objective?


**Reasons To Accept:**

 1.  The utilization of reinforcement learning to improve the performance of the post-processing module is insightful, and the reward function design effectively integrates dialogue-level rewards into the utterance-level rewards.
 2.  GenPPNs offer versatility by applying NLG models with arbitrary structures, making them adaptable and compatible with various systems.


**Reasons To Reject:**

 1. The task settings or baseline methods might not follow common practices. Why do template-based natural language generation (NLG) models outperform other baselines across various metrics?
 2.  The implementation of GenPPNs is based on the Alpaca model with instruction tuning. However, there seems to be a missing ablation experiment that would help determine whether the observed improvement in performance can be attributed to reinforcement learning or instruction tuning.


**Reproducibility:**

4: Could mostly reproduce the results, but there may be some variation because of sample variance or minor variations in their interpretation of the protocol or method.

**Reviewer Confidence:**

4: Quite sure. I tried to check the important points carefully. It's unlikely, though conceivable, that I missed something that should affect my ratings.

---

> ### Author Rebuttal · Authors · 2023-08-29
>
> We are grateful to the reviewer for the thoughtful and encouraging review. We provide explanations and the new results of additional experiments to respond to your concerns and comments as follows.
>
> #### **Q1. The task settings or baseline methods might not follow common practices. Why do template-based natural language generation (NLG) models outperform other baselines?**
> In template-based NLG, high-quality system utterances that optimally represent each dialogue act (DA) are carefully designed manually. For this reason, it is known that template-based NLG performs significantly better than other NLG baselines [1], and the performance reported in this experiment is reasonable.
>
> #### **Q2. There is no study to determine whether the improvement by GenPPN is due to reinforcement learning or instruction tuning.**
> We speculate that the improvement by GenPPN is attributed to reinforcement learning (RL). This is because, as Figure 4 reported in Section A.3 of the appendix, both the reward and Task Success are low at the start of training (i.e., before RL is applied) and improve as RL progresses.
>
> We have yet to investigate the effects of instruction tuning. We plan to validate this by comparing the performance of instruction-tuned models other than Alpaca (e.g., Flan-T5 [2]) and non-instruction-tuned models (e.g., LLaMA [3]).
>
> #### **Q3. In the context of reward using DA contribution, can the value of "e" only be equal to 1 at the end of a dialogue? If this were the case, it raises the possibility of an imbalance between the Successful set and the Failed set.**
> Whether each turn’s “e” value in a given dialogue is 0 or 1 is determined retrospectively based on the outcome at the end of the dialogue, so we believe that the successful set and failed set are unlikely to be  unbalanced. Specifically, for a successful dialogue, “e” is 1 for all turns in that dialogue: $\lbrace(a_t, \hat{a}_t, 1) | t \in T \rbrace$. Conversely, for a failed dialogue, “e” is 0 for all turns in that dialogue: $\lbrace (a_t, \hat{a}_t, 0) | t \in T \rbrace$. Note that we did not consider the ratio of successful and failed dialogues, so there might be some potential for imbalance in that respect.
>
> #### **Q4. Why was GenPPN not applied to GPT-2 w/ RL?**
> We did not apply GenPPN to GPT-2 w/ RL because we wanted to see only how the performance of GenPPN, which considers dialogue-level metrics, differs from NLG models optimized at utterance-level like GPT-2 w/ RL.
>
> Following your review, we applied GenPPN to GPT-2 w/ RL to test its effectiveness as an additional experiment. The results are shown in the following table:
>
>
> | NLG | Success | Inform F1 | Inform Precision | Inform Recall | Book Rate | Turn | DA F1 |
> | :--- | :---: | :---: | :---: | :---: | :---: | :---: | :---: |
> | GPT-2 w/ RL | 72.36 | 76.70 | 73.50 | 85.99 | 76.81 | 7.47 | **81.17** |
> | w/ GenPPN$_\text{mean}$ | 74.02 | 77.10 | 73.87 | 86.82 | 79.19 | 7.54 | 80.98 |
> | w/ GenPPN$_\text{absmax}$ | **75.20** | **78.79** | **75.58** | **88.02** | **79.80** | **7.15** | 80.08 |
>
> From the results, we observed that even for an NLG model optimized at the utterance-level, GenPPN was able to further improve its dialogue performance. However, the utterance-level metric DA F1 slightly decreased from the original GPT-2 w/ RL score, suggesting a limited correlation between dialogue-level and utterance-level performance.
>
> #### **Q5. To what extent did the RL optimization cause GenPPN to over-align with the task completion metrics?**
> Considering the human evaluation results, no significant differences were found in the subjective evaluation scores. Hence, we believe that there is no excessive alignment for task completion and no lack of naturalness. This may be due to the constraint on naturalness using KL divergence, as shown in Equation (7) in Section 3.4, which prevented excessive alignment to task completion only.
>
> #### **Q6. In human evaluations, the user subjectivity assessment metrics deteriorated; is this due to the RL objective?**
> As you pointed out, the slight (but not significant) deterioration in subjective evaluation scores can be attributed to the RL objective, in which measures related primarily to task completion are considered. In fact, in the original PPN paper [4], rewards that only considered task completion did not improve the human subjective evaluation metrics. A similar phenomenon might have occurred in our study.
>
> #### **References:**
> [1] Takanobu et al., 2020. Is Your Goal-Oriented Dialog Model Performing Really Well? Empirical Analysis of System-wise Evaluation. In Proc. SIGDIAL2020, pages 297–310.
>
> [2] Chung et al., 2022. Scaling Instruction-Finetuned Language Models. arXiv preprint arXiv:2210.11416.
>
> [3] Touvron et al., 2023. LLaMA: Open and Efficient Foundation Language Models. arXiv:2302.13971.
>
> [4] Ohashi and Higashinaka, 2022. Post-processing networks: Method for optimizing pipeline task-oriented dialogue systems using reinforcement learning. In Proc. SIGDIAL2022, pages 1–13.

---

### Official Review · Reviewer_LAnm · 2023-08-11

**Soundness:** 4

**Excitement:**

3: Ambivalent: It has merits (e.g., it reports state-of-the-art results, the idea is nice), but there are key weaknesses (e.g., it describes incremental work), and it can significantly benefit from another round of revision. However, I won't object to accepting it if my co-reviewers champion it.

**Paper Topic And Main Contributions:**

The paper describes a module that could be added to a TOD pipeline, particularly to the NLG system or possibly other text-to-text systems in the pipeline, that rewrites its output. The module, GenPPN (generative post-processing network) is optimized using reinforcement learning, where the reward is estimated by the overall task success as well as utterance-level rewards based on which system dialogue acts are preserved in the rewrite and which ones are not. A key highlight of the paper was a heuristic to calculate the reward for recognizing/missing each dialogue act based on its importance for task completion. Results are shown for 3 models (template, SC-LSTM and SC-GPT) on the MultiWoZ benchmark, showing improved task success.

**Reasons To Accept:**

The paper presents an interesting approach to further improve NLG performance in TOD, by adding an RL-tuned network that post-processes the NLG model's output.

That the module is separate means it is somewhat agnostic to a variety of NLG models, increasing its applicability.

The utterance-level reward scheme is clever and could potentially be reused in other TOD work.

Results on MultiWoZ are strong.

The paper is quite clearly written.

**Reasons To Reject:**

It is somewhat hard to motivate this research, since the canonical TOD pipeline is already quite complex, and the motivation for adding another model therein is weak.

Applying the proposed approach, or a variant thereof, directly to the NLG model may be more useful. The use of RL fine-tuning already permits this if the NLG model is a text-to-text LLM.

Viewed vis-a-vis the PPN paper (Ohashi and Higashinaka, 2022), this work may be considered relatively incremental by some.

**Reproducibility:**

4: Could mostly reproduce the results, but there may be some variation because of sample variance or minor variations in their interpretation of the protocol or method.

**Reviewer Confidence:**

4: Quite sure. I tried to check the important points carefully. It's unlikely, though conceivable, that I missed something that should affect my ratings.

---

> ### Author Rebuttal · Authors · 2023-08-29
>
> We are grateful to the reviewer for the thoughtful and encouraging review. We provide discussions and explanations about your concerns as follows.
>
> #### **Q1. Is adding an additional component (i.e., PPN) to the complex TOD pipeline not a drawback?**
> As you pointed out, the post-processing approach may increase complexity. However, we believe that the advantage of optimizing arbitrary modules for the dialogue performance of an entire system will be of more importance.
>
> #### **Q2. Would it not be more effective to fine-tune NLG models directly using reinforcement learning?**
> The primary goal of our study is not to obtain an NLG model with state-of-the-art (SOTA) performance but rather to enhance the performance of NLG irrespective of its architecture or base performance. Therefore, we consider techniques that directly fine-tune the NLG model to be outside the scope of our study.
>
> #### **Q3. Considering the original PPN study [1], this study might seem incremental.**
> We believe our study significantly advances the previous PPN study [1] in two main ways:
> 1. We addressed the challenging but impactful and effective problem of NLG post-processing, which was impossible with conventional PPN.
> 2. We optimized the performance of NLG at the dialogue level for the first time and demonstrated its utility.
>
> #### **References:**
> [1] Ohashi and Higashinaka, 2022. Post-processing networks: Method for optimizing pipeline task-oriented dialogue systems using reinforcement learning. In Proc. SIGDIAL2022, pages 1–13.

---

### Official Review · Reviewer_Bdnj · 2023-08-12

**Soundness:** 4

**Excitement:**

3: Ambivalent: It has merits (e.g., it reports state-of-the-art results, the idea is nice), but there are key weaknesses (e.g., it describes incremental work), and it can significantly benefit from another round of revision. However, I won't object to accepting it if my co-reviewers champion it.

**Paper Topic And Main Contributions:**

This paper introduced a method of post-processing that can be used in the NLG component of a task-oriented dialogue system. This is more challenging for the NLG component than for others (NLU, DST, Policy) as the output is a sequence of tokens rather than one or more slots. The authors use RL-based optimization and a transformer-based generative model to improve the generated output, resulting in improved performance on multiple datasets.

**Questions For The Authors:**

I feel that these same results would be a lot more impactful and important if the authors could show improvements over Template that are not possible without GenPPN.

**Reasons To Accept:**

The paper is very thorough and explains each step of the process clearly. The results are clearly presented and the ablation study is informative. It provides an important final piece of postprocessing for the standard task-oriented dialogue pipeline. The most interesting result is the improvement over the very strong Template baseline.

**Reasons To Reject:**

Looking at Table 2, GenPPN provides good relative improvements for SC-LSTM and SC-GPT. Yet, overall, the absolute performance of these models even with GenPPN does not exceed the performance of Template without GenPPN.  I don't think the authors explain why improving SC-LSTM and SC-GPT is worth doing when it's not better than Template, besides "the lower the performance of the original NLG, the more room there is for improvement by GenPPN", which is true by definition.

With performance worse than Template, I think that SC-LSTM and SC-GPT are weak baselines. The authors show an example in Table 4 where GenPPN helps the SC-LSTM model to produce text for a DA that would be impossible without GenPPN. But, I am curious if
(1) there are similar cases where it improves over Template
(2) the prevalence of such cases
(3) whether it is possible that additional work on the template language can address these.

Without this context, it is very difficult to say whether it is worth all the CPU and GPU cycles that must have been used to produce these results.

**Reproducibility:**

4: Could mostly reproduce the results, but there may be some variation because of sample variance or minor variations in their interpretation of the protocol or method.

**Reviewer Confidence:**

3: Pretty sure, but there's a chance I missed something. Although I have a good feel for this area in general, I did not carefully check the paper's details, e.g., the math, experimental design, or novelty.

---

> ### Author Rebuttal · Authors · 2023-08-29
>
> We are grateful to the reviewer for the thoughtful and encouraging review. We provide explanations and the new results of additional experiments to respond to your concerns and comments as follows.
>
> #### **Q1. Regarding the point that SC-LSTM and SC-GPT, even with GenPPN, cannot surpass the original template NLG. Is there a need to improve SC-LSTM and SC-GPT using GenPPN?**
>
> The primary goal of our study is not to obtain an NLG model with state-of-the-art (SOTA) performance, but rather to enhance the performance of NLG irrespective of its architecture or its base performance. In our experiments, the dialogue performance of NLG models like SC-LSTM and SC-GPT, in addition to Template NLG, were improved using GenPPN, showing that our primary goal has been achieved.
>
> Although NLG baselines in our experiments included Template NLG, known for its high performance [1], in real-world situations, meticulously designed and high-performance NLG models like the template NLG are only sometimes available in other tasks or domains. We believe that GenPPN is a practical approach to improving NLGs in such scenarios.
>
> We also believe there is value in GenPPN when considering objectives other than task completion, such as enhancing naturalness and diversity. As additional experiments, we measured distinct-N of utterances generated by each NLG model during the test. The results are presented in the table below:
>
> | NLG | Task Success | Distinct-1 | Distinct-2 |
> | :--- | :---: | :---: | :---: |
> | Template | 77.25 | 1.91 | 5.37 |
> | Template w/ GenPPN | 78.91 | 1.99 | 5.97 |
> | SC-LSTM | 54.00 | 3.38 | 5.46 |
> | SC-LSTM w/ GenPPN | 72.95 | 3.10 | 7.76 |
> | SC-GPT | 64.94 | 4.39 | 15.89 |
> | SC-GPT w/ GenPPN | 73.63 | 4.62 | 16.08 |
>
> The distinct-N of Template NLG is low, which indicates fixed and mechanical utterances. In contrast, when applying GenPPN to SC-LSTM or SC-GPT, we were able to improve the Task Success score to a level comparable to Template NLG while maintaining the original high distinct-N of the model. This result suggests that we achieved dialogue performance comparable to Template NLG with more diverse utterances.
>
> #### **Q2. To what extent is the help of GenPPN, as confirmed for SC-LSTM, seen for Template NLG?**
> Given that most metrics, including Task Success, have also shown improvements for the Template NLG, it can be inferred that GenPPN was similarly beneficial for the Template NLG as it was for SC-LSTM.
>
> #### **Q3. Can Template NLG be improved by additional work on the template language?**
> By incorporating the insights from the analysis using SC-LSTM into the utterance generation rules of Template NLG, there is potential for improvements in dialogue performance.
>
> #### **References:**
> [1] Takanobu et al., 2020. Is Your Goal-Oriented Dialog Model Performing Really Well? Empirical Analysis of System-wise Evaluation. In Proc. SIGDIAL2020, pages 297–310.

---

### Meta-Review · Area_Chair_me1R · 2023-09-19

**Recommendation:** 4

**Metareview:**

This paper presents GenPPN, a generative post-processing components in dialogue systems to improve task completion. Experiments both through simulation and human evaluation on MultiWOZ dataset show improvement to the task completion performance of TOD system. The work is generally considered thorough, and the paper is well-written and easy to follow. One discussion point during rebuttal is around the high performance baseline Template NLG, which is sufficiently answered by author responses.

---

### Decision · Program_Chairs · 2023-10-07

**Decision:**

Accept-Main

**Comment:**

This paper presents GenPPN, a generative post-processing components in dialogue systems to improve task completion. Experiments both through simulation and human evaluation on MultiWOZ dataset show improvement to the task completion performance of TOD system. The work is generally considered thorough, and the paper is well-written and easy to follow. One discussion point during rebuttal is around the high performance baseline Template NLG, which is sufficiently answered by author responses.